# High-Efficiency Reducing Strain for Producing Selenium Nanoparticles Isolated from Marine Sediment

**DOI:** 10.3390/ijms231911953

**Published:** 2022-10-08

**Authors:** Liying Zhang, Zhuting Li, Lei Zhang, Zhixiao Lei, Liming Jin, Jijuan Cao, Chunshan Quan

**Affiliations:** Key Laboratory of Biotechnology and Bioresources Utilization (Ministry of Education), College of Life Science, Dalian Minzu University, Dalian 116600, China

**Keywords:** marine strains, selenium nanoparticles (SeNPs), extracellular extract, reduction

## Abstract

Selenium nanoparticles (SeNPs) are all important for research because they exhibit a higher degree of absorption and lower toxicity than that of their organic and inorganic forms. At present, there are few reports on marine strains that can reduce Se(IV) to generate Se(0). In this study, a strain that reduces sodium selenite to SeNPs with high efficiency was screened from 40 marine strains. The SeNPs-S produced by the whole cells and SeNPs-E produced by the extracellular extract were characterized by FTIR, UV, Raman, XRD and SEM. Based on the results, the two kinds of SeNPs exhibited obvious differences in morphology, and their surfaces were capped with different biomacromolecules. Due to the difference in shape and surface coating, opposite results were obtained for the antibacterial activity of SeNPs-S and SeNPs-E against Gram-positive and Gram-negative bacteria. Both SeNPs-S and SeNPs-E exhibited no obvious cytotoxicity at concentrations up to 100 μg/mL, but SeNPs-E retained lower cytotoxicity when its concentration increased to 200 μg/mL. This is the first report on the detailed difference between the SeNPs produced by whole cells and cell extracts.

## 1. Introduction

Selenium (Se) is a trace element that is essential for human beings, animals and plants and an important component of human selenoproteins [1]. Excessive selenium can lead to acute toxic reactions, and selenium deficiency is also related to many diseases, including heart, joint and immune system diseases, cancers and diseases involving reproductive functions [2]. In nature, selenium mainly exists in the following valence states: selenate SeO_4_^2−^ (+6), selenite SeO_3_^2−^ (+4), elemental selenium Se^0^ (0) and selenide Se^2−^ (−2) [3]. Among them, selenium oxides (including SeO_4_^2−^ and SeO_3_^2−^) are highly water-soluble, bioaccumulative, and toxic [4]; these oxides exhibit a high toxic effect on the human body and cause serious environmental pollution. Some research has indicated that Se(IV) is the most toxic of all valence states and affects cell respiration and enzyme activity, damages the cell antioxidant system and hinders DNA repair, among other effects [5].

Microorganisms play an important role in the geochemical cycle of selenium. It has been found that many microorganisms can tolerate high concentrations of selenium by either reducing it to selenoproteins, methylating it to selenomethylselenides, which exhibit high volatility, or oxidize low-valence selenium to high-valence selenium through assimilation and dissimilation reduction [6,7,8]. Converting selenium into insoluble and low-toxicity red nanoparticles (SeNPs) is a common dissimilation reduction method that can remove harmful selenium oxygen anions in the environment and produce selenium nanoparticles as potential photocatalysts and anticancer nanoparticles [9,10]. Compared with inorganic selenium and organic selenium, nano selenium exhibits better biocompatibility, lower cytotoxicity and much higher activity [11]. In addition, SeNPs also exhibit good semiconductor, photoconductive, photoelectric, and catalytic properties; thus, SeNPs have wide potential for application in electronic, optic, and life science fields [12].

The common method for synthesizing SeNPs is chemical methods. When reductants were used to reduce SeO_3_^2−^ to SeNPs, dispersants and stabilizers also need to be added, otherwise the products will be aggregated [13]. These two additives may increase the size of SeNPs and result in the loss of some important biological activities. In addition, the chemical regents used in the synthesis process would cause environmental pollution, resulting the difficulties in the future application of SeNPs in the biomedical field [14]. However, the process of biological preparation of SeNPs is simple, rapid, green and pollution-free. As an alternative to chemical and physical methods, more and more attention has been paid to it.

It has been reported that some bacteria, fungi, and actinomycetes have the capability to reduce higher oxidation selenium to produce nano selenium, and among these groups, bacteria are widely studied because they exhibit several advantages, including rapid growth and strong operability [15]. To date, it has been found that a variety of Gram-negative bacteria and Gram-positive bacteria can reduce Se(VI) and Se(IV) to synthesize SeNPs. Oremland et al. found that three different selenite and selenite respiratory bacteria (*Sulfurospirium barnesii*, *Bacillus selenium*, *Selenihalanaewbacter shriftii*) could synthesize stable and uniform extracellular Se nanoparticles with a diameter of approximately 300 nm [16]. Dhanjal and Cameotra identified a strain of *Bacillus cereus* that can reduce selenite under aerobic conditions to generate red Se nanoparticles with a diameter of 150–200 nm [17]. Moreover, probiotics can also reduce selenite to generate SeNPs. Eszenyi et al. showed that using sodium selenite (NaHSeO_3_) as the selenium source and adding probiotics, such as *Lactobacillus* sp., *Bifidobacterium* sp. and *Streptococcus thermophilus*, during the fermentation process can also result in SeNPs that are 100 to 500 nm in diameter [18]. Compared with bacteria, it is easier to control fungi in the reactor because fungi can better withstand adverse environments, such as shock and fluid pressure. Fungi can also secrete abundant extracellular proteins, polypeptides and secondary metabolites. In addition, the nanoparticles it produces are easy to separate, and the downstream treatment process is simple [19,20]. However, studies on the synthesis of SeNPs by fungi occurred slightly later than those by bacteria, and the currently available resources on fungi remain very limited. Zare et al. isolated a strain of *Aspergillus terreus* from soil samples and added Se(IV) to its culture medium supernatant to synthesize SeNPs [21]. Vetchinkina et al. found that the edible basidiomycete *Lentinula edodes* can reduce Se(IV) and accumulate SeNPs in the mycelium [22]. Research on actinomycetes was recently performed, and the strains that were reported to synthesize SeNPs, such as *S. bikiniensis* strain Ess_amA-1 and *S. microflavus* strain FSHJ31, all belonged to *Streptomyces* [23,24]. Aside from microorganisms, SeNPs have been synthesized using different plants or their extracts, including tarragon extract, ginger extract, fenugreek seed extract, *Clausena dentata* leaf aqueous extract, and *Aloe vera* leaf extract [25,26,27,28,29].

The particularity of the marine environment and the complex ecological functions of marine species endow marine microorganisms with metabolic pathways and adaptive mechanisms that differ from those of terrestrial microorganisms [30]. Marine microorganisms have become an important source of new drugs due to their abundant secondary metabolites, which exhibit unique structures and significant biological activity [31]. To date, there are only a few reports about marine strains that can reduce Se(IV) to SeNPs, and these reports include *B. amyloliquefaciens* SRB04, which was isolated from the coast of the Caspian Sea; *Sargassum fusiforme*, which was distributed in the shallow sea areas of some provinces in China; and *Yarrowia lipolytica* and *Exiguobacterium aestuarii* SBG4 MH185868, which were isolated from marine water in India [32,33,34,35]. The focus of these studies mainly involved regulating the output, production cost and biological activity of SeNPs. However, these studies did not determine the cellular part in marine bacteria at which the reduction in Se(IV) to Se(0) is completed, and the difference in SeNPs produced by the whole cells and the extraction was not examined.

In this study, by investigating the tolerance and reduction ability of Na_2_SeO_3_, a strain with an ability to perform high-efficiency reduction to produce SeNPs was successfully screened from 40 strains that were isolated from marine sediment in the Arctic Ocean. The SeNPs produced by the whole cells and the extracellular extract were characterized by Fourier transform infrared spectroscopy (FIIR), ultraviolet spectroscopy (UV), Raman spectroscopy (Raman), X-ray diffraction (XRD) and scanning electron microscopy (SEM). In addition, the antibacterial activity and cytotoxicity of SeNPs were evaluated. We compared the properties of SeNPs produced by different reduction methods for the first time; based on these results, the possibility of large-scale and rapid synthesis of nano selenium using bacterial extract is discussed.

## 2. Results and Discussion

### 2.1. Isolation and Characterization of a High-Efficiency Reducing Strain for Producing SeNPs

As shown in Figure 1, forty strains isolated from Arctic Ocean sediments were streaked on LB agar plates containing Na_2_SeO_3_. According to their growth and color changes, the strains capable of reducing SeO_3_^2−^ to produce SeNPs were preliminarily screened. The results showed that 30 strains changed to brick red at different depths (Appendix A). Then, the strains were added to LB liquid medium containing 5 mM Na_2_SeO_3_ to rescreen by measuring cell growth. After 48 h of culture, it was found that 12 strains (2, 8, 21, 33, 37, 45, 53, 66, 67, 71, 72 and 76) grew well, and their OD_600_ values were greater than 2.20 (Appendix A). Therefore, we selected these 12 strains for subsequent screening. Next, the 12 rescreened strains were inoculated into LB medium containing Na_2_SeO_3_ at different concentrations. After 24 h of culture at 33 °C and 180 rpm, strains 72, 33, and 21 exhibited a deeper color at a concentration of 80 mM. We inferred that these three strains had a stronger ability to reduce Na_2_SeO_3_ compared to that of the strains. Therefore, strains 72, 33, and 21 were selected for further experiments.

The growth of strains 72, 33, and 21 in the presence of 5 mM Na_2_SeO_3_ and their reduction in Na_2_SeO_3_ are shown in Figure 2. Figure 2a–c shows that the process of reducing Na_2_SeO_3_ by these three strains mainly occurred in the logarithmic growth phase. After 6 h of culture, the bacterial solution of strain 33 turned red, indicating that SeNPs was produced at this time. The culture solutions of strain 21 and strain 72 turned red after 8 h of culture. When cultured for 4 h, the concentration of Na_2_SeO_3_ in the three culture solutions was lower than 50% of the initial concentration, and the synthesis of SeNPs obviously lagged behind the consumption of SeO_3_^2−^. The reasons for this phenomenon are that the dissimilation reduction in SeNPs is mainly divided into the following steps: transport, reduction, and assembly. The decrease in the concentration of SeO_3_^2−^ in the first 6 h solution is due to the transport of extracellular SeO_3_^2−^ into the cell, which reacts with intracellular glutathione to generate GS-Se-SG and is further reduced to GS-Se^−^ under the action of glutathione reductase (GR) or thioredoxin reductase (TR). GS-Se^−^ is unstable and generates GSH and Se^0^ through hydrolysis. Therefore, the generation of Se^0^ lags behind the reduction in SeO_3_^2−^ [36]. Furthermore, the relevant literature indicates that bacteria can directly convert SeO_3_^2−^ into selenium-containing proteins through assimilation reduction, which is another reason why the consumption rate of SeO_3_^2−^ is higher than the generation rate of SeNPs [37].

To compare the ability of strains 72, 33 and 21 to reduce SeO_3_^2−^, their rate of SeO_3_^2−^ reduction was fitted (Table 1). After calculations were performed, it was found that the reduction reaction of the three strains conformed to the first-order kinetic equation. Figure 2d,e shows that the reaction time has an obvious linear relationship with the SeO_3_^2^ concentration (ln (C/C_0__)_), and the correlation coefficients R^2^ are 0.963, 0.973, and 0.953, respectively. We also calculated the reaction rate constant, half-life and SeO_3_^2−^ reduction rate to compare the reducing ability of these three strains more clearly. The rate constants of 71, 33, and 22 strains for SeO_3_^2−^ reduction were 0.1081, 0.1297, and 0.1162, and the half-lives were 6.41, 5.34, and 5.96 h^−1^, respectively, indicating that the above three strains all exhibited high reduction rates. The SeO_3_^2−^ reduction rates of the three strains were 93.7%, 96.7%, and 95.2% within 24 h.

The three strains can all grow in the presence of higher Na_2_SeO_3_ concentrations (>100 mM). Compared with some strains reported [38,39,40], these three strains not only exhibit high Na_2_SeO_3_ tolerance but also have a higher reduction speed and reduction rate. Among these strains, strain 33 is the most efficient.

16S rRNA gene sequencing was used to characterize the marine strain 33 that was screened. After 24 h of incubation on LB medium, as shown in Figure 3a, the colonies were milky white and opaque; the colonies exhibited folds at the edges, shrinking, and bulging toward the middle; and mucinous secretions were present when the bacteria were picked up. Analysis of the 16S rDNA gene sequence from strain 33 revealed 99% identity with *Bacillus species*. The 16S rDNA gene sequence was retrieved by NCBI and compared with its 16S rDNA sequence, and the phylogenetic tree was calculated using Clustal ver. 1.83 and MEGA software ver. 11, and the results are shown in Figure 3b. Based on the above results, we conclude that strain 33 belongs to *Bacillus* sp., and named it *Bacillus* sp. Q33.

### 2.2. The Reduction Location of SeO_3_^2−^ in Bacillus sp. Q33

It has been reported that bacteria can reduce SeO_3_^2−^ to produce SeNPs in different parts of the cell, including intracellular, extracellular, intracellular and extracellular parts, between the cell wall and the plasma membrane [9,16,41,42]. To determine the location of SeO_3_^2−^ reduction in *Bacillus* sp. Q33, three kinds of extraction solutions, including extracellular, pericellular, and intracellular solutions, were extracted. A time-dependent color change was observed in the extracellular extract after incubation at 33 °C for 24 h, as shown in Figure 4. The initial light-yellow color of the solution gradually changed to red with time. After incubation for 12 h, the solution turned dark red, indicating that a large amount of SeNPs was produced. However, there was no obvious discoloration in the pericellular and intracellular extracts. This indicated that the extracellular extraction solution contains key proteins and reducing substances that can reduce SeO_3_^2−^, which causes the solution to turn red. However, the pericellular and intracellular extraction may lack or do not contain reducing active substances, which hinders the occurrence of electron transfer. Therefore, the reduction reaction is not complete, and the color of the solution does not change.

### 2.3. Synthesis and Characterization of SeNPs

Biosynthesis is a facile, safe, biocompatible, eco-friendly and recyclable method of preparing SeNPs. The properties, size and morphology of nanoparticles can be easily controlled by changing the incubation temperature, pH, reaction time, metal ion concentrations and quantity of organic material [12]. However, the process of separating biological reduction to produce SeNPs is cumbersome, as it necessitates crushing and multiple centrifugations, and completely removing the cell debris and components is difficult. When the extraction solution is used for reduction, only a simple centrifugal cleaning is needed in the separation process because the bacterial cells do not directly participate, and the preparation process is more convenient. The reduction in SeO_3_^2−^ by *Bacillus* sp. Q33 occurs outside the cell, so the extracellular extract can be used to directly synthesize SeNPs. However, there is no detailed report on the difference between SeNPs synthesized by whole-cell reduction and extracellular extract reduction. Based on this, we synthesized SeNPs by two methods, in which the production by whole-cell reduction was recorded as SeNPs-S and the extracellular extract was recorded as SeNPs-E, and compared their properties in detail.

Figure 5a shows the FTIR spectra of SeNPs and SeNPs-E. The functional groups are basically the same, and they all exhibit obvious characteristic absorption peaks at ~3400 cm^−1^, ~2930 cm^−1^, ~1640 cm^−1^, ~1520 cm^−1^, ~1400 cm^−1^ and ~1100 cm^−1^. The absorption peak at ~3300 cm^−1^ is produced by O-H stretching vibration in protein; the 2930 cm^−1^ absorption peak is generated by the stretching vibration of -C-H- in -CH_2_-; 1640 cm^−1^ is the carbonyl stretching vibration in amide, 1520 cm^−1^ is the absorption peak of N=N double bond; 1400 cm^−1^ absorption peak is -CH- stretching variation and the absorption peak at 1100 cm^−1^ is the C-N stretching vibration absorption. Amino and hydroxyl groups were attached to the surface of SeNPs that were produced by both methods. In addition, SeNPs-E has an absorption peak at ~2853 cm^−1^, and the peak between 2800–3000 cm^−1^ is more prominent than that of SeNPs-S. At the same time, the characteristic absorption peak of -CH- in -CH_3_ appears at 1453 cm^−1^, indicating that the biomacromolecules capped on its surface contain a high proportion of Leu, Ile, Val and other amino acids with -CH_3_ groups. However, SeNPs-S has a characteristic absorption peak at 748 cm^−1^, which belongs to the external bending vibration absorption of the substituted benzene ring. Therefore, the biomacromolecules on its surface may contain Phe, Tyr, Trp and other amino acid residues with substituted benzene rings. The obtained findings verified that some differences occur in the surface coverings of the SeNPs prepared by whole cells and their extracellular extracts, which may lead to some differences in their biological activities.

UV spectroscopy was used to monitor the biosynthesis of SeNPs-S and SeNPs-E, which exhibited wavelengths between 200 and 400 nm. The broad peak recorded in the range 250–300 nm, as represented in Figure 5b, was assigned to the surface plasmon vibrations of SeNPs-E, which was recognized as a characteristic peak of SeNPs, as previously reported elsewhere [43,44]. While the SeNPs-S that were reduced by the whole cells only showed a small turning in this region, there was a certain difference in the optical properties of SeNP synthesis by the two methods.

The patterns of XRD analysis are shown in Figure 5c, which revealed whether SeNPs-E and SeNPs-S contain amorphous or crystalline structures. The XRD pattern of SeNPs-S revealed a broad peak at 23.5° and a sharp peak at 30° at 2*θ* values in the spectrum, while these two peaks observed in the spectrum of SeNPs-E were strong and sharp. These two peaks accounted for the lattice planes of 100 and 101, respectively, showing the presence of crystalline Se in the nanoparticle structure [45]. There are also a few sharp peaks in the XRD pattern of SeNPs-E, which were observed at 41.2°, 43.6°, 45.4°, 51.7°, 55.9°, 61.5° and 65.4°. These peaks corresponded to the Miller indices (1 1 0), (1 0 2), (1 1 1), (2 0 1) and (2 0 2), respectively, in line with those of Se(0) on the standard data card (JCPDS card No. 06-0362) [46]. However, these peaks were not significant in SeNPs-S, indicating that the SeNPs produced by whole-cell reduction are spherical or may be a small amount of cell debris and that other impurities are mixed in the separation process.

The Raman spectra of the two selenium nanoparticles are shown in Figure 5d. It has been shown that the Raman spectrum of SeNPs is generally divided into the following signal peaks: one peak is at approximately 234 cm^−1^, corresponding to rod-loaded selenium (t-Se), and the other peak is at approximately 255 cm^−1^, corresponding to spherical selenium (m-Se) [47]. Figure 5d shows that SeNPs-E has a signal peak at 235.4 cm^−1^, indicating the presence of rod-loaded nano-Se. SeNPs-S showed a signal peak at 249.8 cm^−1^, indicating the presence of spherical SeNPs.

Figure 6 is the morphology and composition characterization of SeNPs-E and SeNPs-S. As shown in Figure 6a, *Bacillus* sp. Q33 has a short rod with a length of approximately 2 µm and is surrounded by many white spherical nanoparticles. Elemental analysis data obtained from energy dispersive spectra confirmed the presence of SeNPs. The signal for Si was derived from the silicon wafer that was used in sample preparation. Signals for C and O are typical of biological systems (Figure 6b). SeNPs-S was spherical, with an average particle size of 159.2 nm, and its zeta potential was −42.0 ± 0.4 MV (Figure 6c,d). Recently, the particle size of SeNPs produced by marine strains was reported to be less than 100 nm [32]. Under the existing experimental conditions, the particle size of SeNPs synthesized by *Bacillus* sp. Q33 is slightly larger. Its zeta potential was similar to that of SeNP synthesis by the marine strain *E. aestuarii* SBG4 MH185868 at pH 6.5 [35]. SeNPs-E is close to a tetragonal body with an uneven surface, with an average particle size of 218.6 nm and zeta potential of −40.9 ± 0.6 MV (Figure 6e,f). The SEM results further confirmed that some differences were observed in the structure and morphology of SeNPs prepared by the two reduction methods. Most of the SeNPs produced by microbial reduction are spherical, although there are few reports on other shapes. In terms of bacteria, strain *Z. ramigera* can produce Se nanorods, which are known as trigonal selenium (t-se), after 48 h of incubation with 3 mM solid selenium [48]. Strain *S. bikiniensis Ess_amA-1* can reduce SeO_2_ to selenium nanorods (SeNrs) [23]. This shape is produced due to the higher free energy of Se nanospheres, which subsequently deposited on the surface of t-Se and acted as seeds for the growth of uniform t-Se nanorods. The process of transforming small SeNPs into large ones was in agreement with the Ostwald ripening process [49]. Pallavee et al. found that haloarchaeon *Halococcus salifodinae* BK18 can produce rod-shaped nanoparticles with an average length of 129 nm, an average diameter of 10 nm, and an aspect ratio of 13:1 after 7 days of coculture with Na_2_SeO_3_ [50]. Diko reduced sodium selenite by *Trichoderma* sp. WL-Go culture broth to produce spherical and pseudo spherical SeNPs [51]. A novel yeast Magnusiomyces ingens can also produce spherical and pseudospherical Se particles [52]. We report for the first time that the extracellular extract of *Bacillus* sp. Q33 isolated from marine sediment can synthesize SeNPs with a tetragonal shape. At present, the particle size distribution of nano-Se is wide. In the future, we will optimize the synthesis conditions to improve the quality of SeNPs.

### 2.4. Antibacterial Activity of Selenium Nanoparticles

Figure 7 shows the inhibitory activities of SeNPs-E and SeNPs-S against four pathogenic bacteria, two Gram-negative bacteria *E. coli* and *P. aeruginosa*, and two Gram-positive bacteria *S. aureus* and *L. monocytogenes*. Taking *P. aeruginosa* as an example (Figure 7a,b), after 4 h of incubating pathogenic bacteria with different concentrations of SeNPs at 37 °C, the bacterial suspensions were diluted 10^−1^, 10^−2^, 10^−3^ and 10^−4^ times with water, and then 2.5 μL of these solutions was transferred to agar plates with LB. The agar plates were incubated overnight at 37 °C, and then the bacterial colony-forming units were observed and counted. The inhibition of bacterial growth by different concentrations of SeNPs-E and SeNPs-S is shown in Figure 7c–f. The SeNPs prepared by the two reduction methods all showed inhibitory activity against four pathogens, and with the increase in concentration, the growth of bacteria was obviously inhibited. For Gram-negative bacteria, the inhibitory ability of SeNPs-S is better than that of SeNPs-E, while that for Gram-positive bacteria is the opposite. Taking the concentration of 200 μg/mL as an example (Table 2), SeNPs-E showed the strongest inhibitory ability against *S. aureus*, with an inhibitory rate of 96.8%. The difference between the antibacterial properties of SeNPs-S and SeNPs-E may be related to the different proteins loaded on their surface. The detailed rationale needs to be further studied.

### 2.5. Cytotoxicity of Selenium Nanoparticles

The cell viability of HCoEpiC exposed to different concentrations of SeNPs-S and SeNPs-E was measured using 0.02% Triton × 100 as a positive control to test SeNP cytotoxicity. As shown in Figure 8, SeNPs-S and SeNPs-E exhibited no obvious cytotoxicity at concentrations up to 100 μg/mL, and the effect of the two kinds of SeNPs on the proliferation of HCoEpiC cells was significantly less than that of the positive control. According to ISO10993-5 [53], material that reduces cell viability below 70% of the negative control viability is considered to be potentially cytotoxic, so the effects of both SeNPs-S and SeNPs-E are not classified as cytotoxic at these doses. Furthermore, when the concentration increased to 200 μg/mL, the cell viability of the SeNPs-S system decreased to 64.2% of the negative control, while that of SeNPs-E remained at 86.9%, indicating that the cytotoxicity of SeNPs-S was higher than that of SeNPs-E at this concentration. This may be due to the presence of a small amount of bacterial residues during the separation of SeNPs-S, which are toxic to cells. However, the preparation process of SeNPs-E does not involve the participation of bacteria, its separation is more convenient, and the purity of SeNPs is higher; thus, SeNPs-E still shows low toxicity to HCoEpiC cells at 200 μg/mL.

## 3. Conclusions

In this study, a strain with high efficiency in reducing sodium selenite to SeNPs was screened from 40 marine-derived strains through primary screening, rescreening, tolerance, and the reduction ability of SeO_3_^2−^. The strain was identified by 16S rRNA gene sequencing and named *Bacillus* sp. Q33. The reduction in SeO_3_^2−^ by this strain mainly occurred extracellularly. Two SeNPs, SeNPs-S and SeNPs-E, were synthesized by whole cells and extracellular extract and characterized by FTIR, UV, Raman, XRD, and SEM. The results showed that the SeNPs-S reduced by the whole cell was spherical, while the SeNPs-E reduced by the extracellular extract was mainly tetragonal, and its particle size was larger than that of SeNPs-S. These two kinds of SeNPs are coated with biomacromolecules, but the kinds of amino acid residues in them are different. The results of antibacterial experiments showed that SeNPs-S and SeNPs-E exhibited obvious inhibitory effects on the four pathogenic bacteria. Due to the difference in shape and surface coating protein, the antibacterial activity of SeNPs-S against Gram-negative bacteria was higher than that against Gram-positive bacteria, especially against *S. aureus*. When the concentration was lower than 100 μg/L, the two kinds of SeNPs all showed lower cytotoxicity to HCoEpiC cells.

Here, we compared the difference in SeNPs that were produced by whole cells and extracellular cell extracts for the first time in detail. The results showed that the Se nanoparticles prepared by the reduction in the cell extracellular extracts exhibited lower cytotoxicity and showed obvious inhibition against four pathogenic bacteria. These results indicated that the SeNPs produced by the cell extract also had good biological activity indicating its potential application in the field of biomedicine. Next, we will optimize the reducing process in the extracellular extract to prepare SeNPs with more uniform size. In addition, a detailed study of the biomacromolecules coated on the SeNPs and the key proteins involved in the reduction in sodium selenite in the extracellular extract and the whole cells will be performed.

## 4. Materials and Methods

### 4.1. Materials

Forty marine-derived strains were isolated from the sediment of the Arctic Ocean. *Pseudomonas aeruginosa*, *Escherichia coli*, *Staphylococcus aureus*, *Listeria monocytogenes* were stored in our lab. Yeast powder and beef extract peptone were purchased from Aobox (Beijing, China). Agar was purchased from Sangon Biotech (Shanghai, China). Absolute ethanol, acetone, potassium bromide, sucrose, ascorbic acid, hydrochloric acid, tris(hydroxymethyl)aminomethane hydrochloride (TRIS), were all purchased from Kemiou (Tianjin, China). Analytical-grade Na_2_SeO_3_ was purchased from Alfa Aesar (Shanghai, China).

### 4.2. Screening and Identification of High-Efficiency Reducing Strains for Producing Se Nanoparticles

#### 4.2.1. Screening of Strains Capable of Reducing Sodium Selenite

Forty marine-derived strains, which were stored at −80 °C, were thawed, inoculated on LB agar plates, and cultured at 33 °C for 24 h. Single colonies were picked out, placed in LB liquid medium and cultured at 33 °C and 180 rpm for 24 h. The activated strains were inoculated into solid medium containing Na_2_SeO_3_ and were cultured at 33 °C for 72 h to observe the colony color. The single red-colored colony was then subcultured on LB liquid medium containing 1 mM Na_2_SeO_3_ for 24 h at 33 °C and 180 rpm to select high-efficiency reducing strains by determining the OD_600_.

#### 4.2.2. Sodium Selenite Tolerance of the Strain

Through primary screening and rescreening, 12 strains were screened out from 40 marine strains. For the Na_2_SeO_3_ tolerance test, 150 µL bacterial culture was added to a certain volume of LB liquid medium, and then sterilized Na_2_SeO_3_ solution was added so that the final concentrations of the system were 10, 20, 40, 60, 80, 100, 200, 300, 400 and 500 mM. After mixing, the mixture was incubated for 24 h at 33 °C and 180 rpm, and the color change of the solution was detected.

#### 4.2.3. Reducing Rate of Sodium Selenite by the Strain

The Na_2_SeO_3_ reducing ability was evaluated using the method of Li et al. [54]. In the selenite reduction tests, cells were added to 100 mL of LB medium with a final OD_600_ of 1.0. The reduction tests were carried out at 33 °C, and the reduction efficiency was expressed by the decrease in selenite concentration. To quantify the selenite concentration, a subsample at each time point was centrifuged to remove cells and Se(0) particles. The supernatant (2 mL) was mixed with 1 mL of hydrochloric acid (4 mol/L) and 1 mL of ascorbic acid (1 mol/L) with vortexing. The selenite concentration was measured spectrophotometrically at 500 nm after 10 min.

#### 4.2.4. Strain Identification

The strain to be tested was inoculated in LB liquid medium at an inoculation amount of 1% and cultured at 33 °C and 180 rpm for 24 h. The bacterial DNA extract was used as a template, and the 16S rDNA sequences were amplified by polymerase chain reaction (PCR) using the universal primers 8F (5′-AGAGTTTGATCCTGGCTCAG-3′) and 1492R (5′-TACGGCTACCTTGTTAGGACT-3′). PCR was carried out with one cycle of heat treatment at 95 °C for 5 min, a total of 32 cycles of denaturation at 94 °C for 30 s, annealing at 50 °C for 30 s, extension at 72 °C for 1 min, and a final extension at 72 °C for 10 min. The PCR products were stored at 4 °C and later analyzed by 1% agarose gel electrophoresis. Sequencing of the PCR products was performed by Sangon Co. Ltd. (Shanghai, China). The sequences were submitted to GenBank for homology searches using BLAST (http//ncbi.nim.nih.gov, accessed on 30 August 2022). The sequences of 16S rDNA were aligned with those of related bacterial strains retrieved from GenBank using CLUSTAL X. A phylogenetic tree was constructed using MEGA 11.

### 4.3. Determination of SeO_3_^2−^ Reducing Location in Bacillus sp. Q33

Ten milliliters of the bacterial solution that was cultured for 24 h was centrifuged at 8000 rpm for 10 min to obtain the supernatant. After the precipitate was washed with precooled ultrapure water, 10 mL of Tris-HCl buffer at pH 8 was added to resuspend the precipitate, which was then centrifuged at 8000 rpm for 10 min to obtain the supernatant. Next, 10 mL of 25% sucrose solution was used to resuspend the precipitate, which was centrifuged at 8000 rpm for 10 min to obtain the supernatant. The supernatant obtained in the above three steps was mixed and recorded as the extracellular extract. After washing the remaining precipitate with precooled ultrapure water, 10 mL of precooled ultrapure water was added to resuspend the precipitate, which was then placed into an ice water bath, shaken for 10 min, and centrifuged at 12,000 rpm for 10 min. The supernatant was the pericellular extract. A total of 10 mL of Tris HCl buffer at pH 8 was added to resuspend the precipitate, the solution was centrifuged for 10 min at 8000 rpm, the supernatant was placed into a small beaker then crushed for 10 min with an ultrasonic cell breaker, the crushing solution was centrifuged for 10 min at 12,000 rpm, and the intracellular extract was obtained.

A certain amount of Na_2_SeO_3_ solution was added to 300 mL of the extracellular extract, pericellular extract and the intracellular extract to a final concentration of 5 mM. The mixture was incubated at 33 °C and 180 rpm for 48 h, and the formation of SeNPs was monitored.

### 4.4. Synthesis and Characterization of SeNPs

#### 4.4.1. Synthesis of SeNPs-S by Whole Cells of *Bacillus* sp. Q33

Cells cultivated in 300 mL of LB liquid medium containing 5 mM Na_2_SeO_3_ at 33 °C for 48 h were sonicated (working time 5 s, interval 3 s, total time 30 min) and centrifuged at 5000 rpm for 5 min. The cell debris was discarded, and the supernatant was centrifuged at 12,000 rpm for 10 min. The supernatant was discarded, and the precipitate was washed with ultrapure water 3 times. After the waste liquid was discarded, the supernatant was washed with absolute ethanol 3 times and vacuum dried at 60 °C for 24 h. This sample is named SeNPs-S.

#### 4.4.2. Synthesis of SeNPs-E by the Extracellular Extract of *Bacillus* sp. Q33

A certain amount of Na_2_SeO_3_ was added to 300 mL of L1 solution to make the final concentration of the system 5 mM. The mixture was incubated at 33 °C and 180 rpm for 48 h. Then, the mixture was centrifuged for 15 min at 10,000 rpm, the supernatant was discarded, the precipitate was washed twice with ultrapure water, absolute ethanol was added to wash the precipitate twice after discarding the waste liquid, and the mixture was vacuum dried at 60 °C for 24 h. This sample is named SeNPs-E.

#### 4.4.3. Characterization of SeNPs-S and SeNPs-E

The crystallographic structures of SeNPs-S and SeNPs-E were characterized by a Shimadzu XRD-600 (Kyoto, Japan) X-ray diffractometer with Cu Kα1 radiation (λ = 0.15406 nm). Field-emission scanning electron microscopy (Hitachi, Tokyo, Japan) was used to inspect the morphologies and structures of the samples. The UV—vis absorption spectra of SeNPs-S and SeNPs-E were recorded by a Lambda 750 spectrophotometer (Perkin-Elmer, Waltham, MA, USA). The infrared spectrum and Raman spectrum of the two samples were recorded by an infrared spectrometer (SHIMADZU, Kyoto, Japan) and Raman spectrometer (HORIBA, Kyoto, Japan). The Z-potential and dynamic light scattering (DLS) measurements were carried out using a Zetasizer-Nano ZS from HORIBA Instruments (Kyoto, Japan). DLS characterization was performed on SeNPs-S and SeNPs-E dispersed in H_2_O.

### 4.5. Biological Activity of SeNPs

#### 4.5.1. Antibacterial Activity of SeNPs

Colony-forming unit (CFU) assays using *E. coli*, *S. aureus*, *L. monocytogenes and P. aeruginosa* (reserved in our laboratory) were performed to test the antibacterial activity of SeNPs-S and SeNPs-E. First, a single colony of four pathogens was taken from an agar plate, inoculated into 20 mL of LB and cultured overnight at 37 °C. Then, 200 µL of the overnight culture solution was transferred into 10 mL fresh LB and cultured for 4 h at 37 °C. One hundred microliters of sterile water with different concentrations of SeNPs-S or SeNPs-E particles was added to each well of 96-well plates. Then, 100 µL of culture solution was added to each well. After 4 h of incubation at 37 °C, the bacterial suspensions were diluted 10^−1^, 10^−2^, 10^−3^ and 10^−4^ times with water, and then 2.5 μL of these solutions was transferred to agar plates with LB. The agar plates were incubated overnight at 37 °C, and then the bacterial colony-forming units were observed and counted.

The method of calculating the strain growth rate is as follows:Growth rate (%) = CFU _Experimental group_/CFU_Control_Inhibitory rate (%) = (CFU _Control_ − CFU _Experimental group_)/CFU_Control_

#### 4.5.2. Cytotoxicity of SeNPs

HCoEpiC cells in logarithmic growth phase were taken and inoculated in 96-well culture plates with 1.2 × 10^4^ cells per well. The culture plate was placed into a constant temperature incubator with 5% CO_2_, saturated humidity and 37 °C temperature for 12 h. Then, the old culture medium was removed, and 100 µL of 0, 25, 50, 100 and 200 μg/mL SeNPs-S and SeNPs-E dissolved in complete culture medium was added to each well. Triton × 100 (0.02%) was used as the positive control. After 24 h of coculture, the old medium was discarded, and 100 μL MTT (5 mg/mL) and cell culture medium (1:9) were added and incubated for 4 h. Subsequently, the supernatant was discarded, 100 µL DMSO was added to each well in the dark, the solution was shaken at room temperature for 10 min to completely dissolve the purple crystals, and a multifunctional enzyme labeling instrument was used to measure the OD_490_.

## Figures and Tables

**Figure 1 ijms-23-11953-f001:**
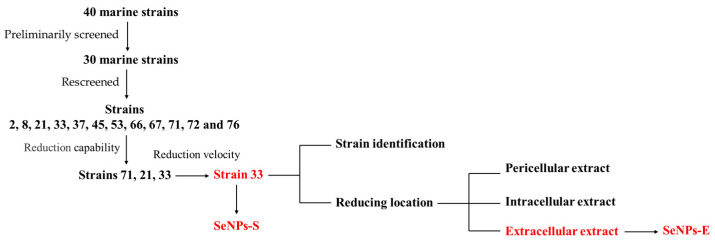
Flowchart for methodology adopted in this study.

**Figure 2 ijms-23-11953-f002:**
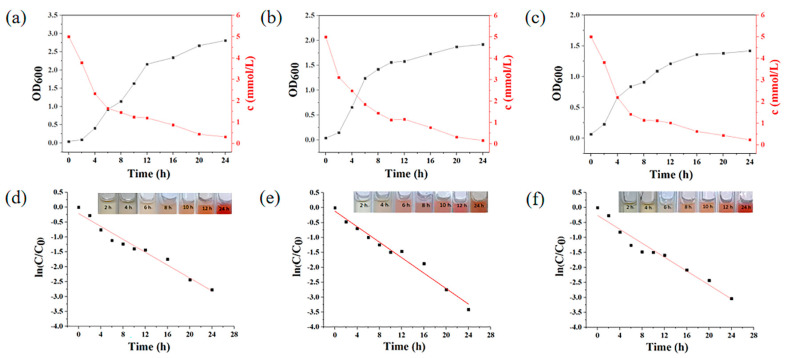
Time course of growth and SeO_3_^2−^ removal by three strains in LB medium supplemented with 5 mM NaSeO_3_. (**a**) Strain 72. (**b**) Strain 33. (**c**) Strain 21. The kinetic fitting curve of reduction reaction and color change of reaction system. (**d**) Strain 72. (**e**) Strain 33. (**f**) Strain 21.

**Figure 3 ijms-23-11953-f003:**
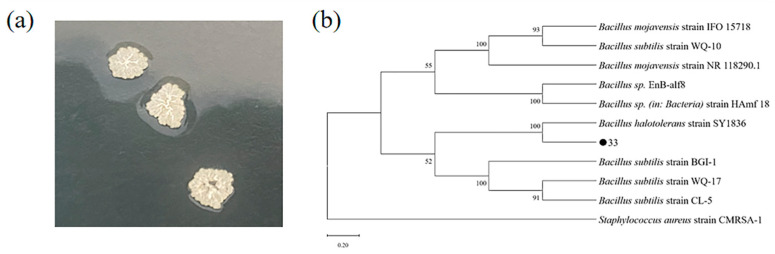
Morphological (**a**) and phylogenetic tree (**b**) of Strain 33.

**Figure 4 ijms-23-11953-f004:**
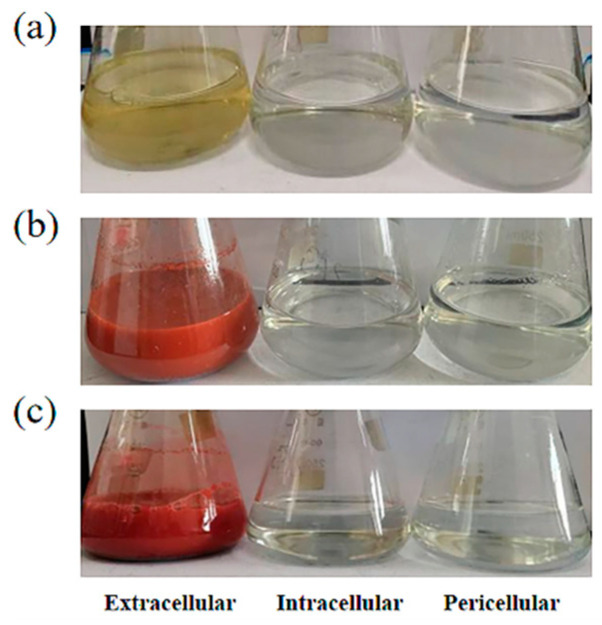
Reduction in SeO_3_^2−^ by extracellular, pericellular, and intracellular extracts of *Bacillus* sp. Q33 containing 5 mM Na_2_SeO_3_. (**a**) 4 h (**b**) 12 h (**c**) 24 h.

**Figure 5 ijms-23-11953-f005:**
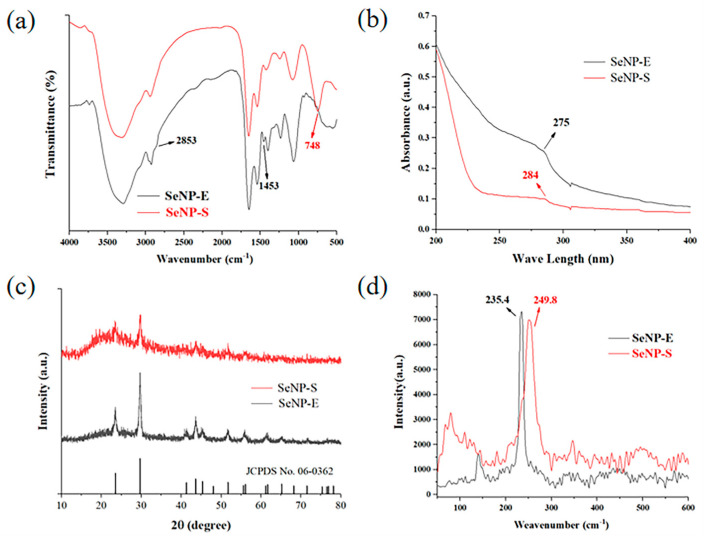
Spectroscopic characterization of SeNPs−E and SeNPs−S. (**a**) Infrared spectra. (**b**) UV spectra. (**c**) XRD diffraction patterns. (**d**) Raman spectra.

**Figure 6 ijms-23-11953-f006:**
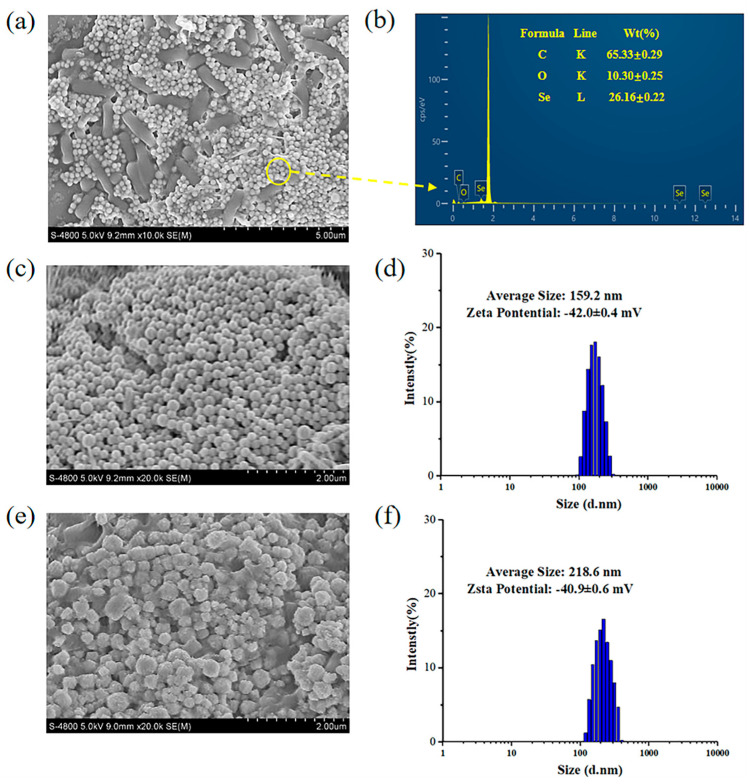
Morphology and composition characterization of SeNPs−E and SeNPs−S. (**a**) SEM micrographs of *Bacillus* sp. Q33 and SeNPs−S. (**b**) Energy dispersive X−ray diffraction. (**c**) SEM micrographs of SeNPs−S. (**d**) Particle size distribution of SeNPs−S. (**e**) SEM micrographs of SeNPs−E. (**f**) Particle size distribution of SeNPs−E.

**Figure 7 ijms-23-11953-f007:**
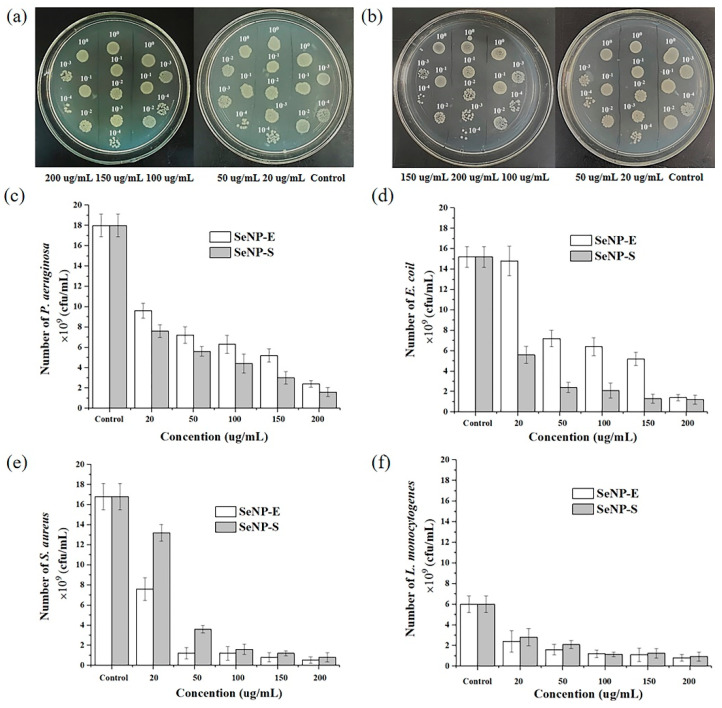
Colony−forming units (CFU) assay using four pathogens after treatment with SeNPs−S and SeNPs−E with varying concentrations from 20 μg/mL to 200 μg/mL. (**a**,**b**) Agar plate images of CFU test of *P. aeruginosa.* The original (bacteria + SeNPs) solution is 10^0^, while 10^−1^, 10^−2^, 10^−3^ and 10^−4^ mean diluting the original solution 10, 100, 1000 and 10,000 times, respectively, to make the colonies more countable. Viable cell concentration of *P. aeruginosa* (**c**), *E. coli* (**d**), *S. aureus* (**e**) *and L. monocytogenes* (**f**) after 24 h exposure to different concentrations of SeNPs-S and SeNPs-E.

**Figure 8 ijms-23-11953-f008:**
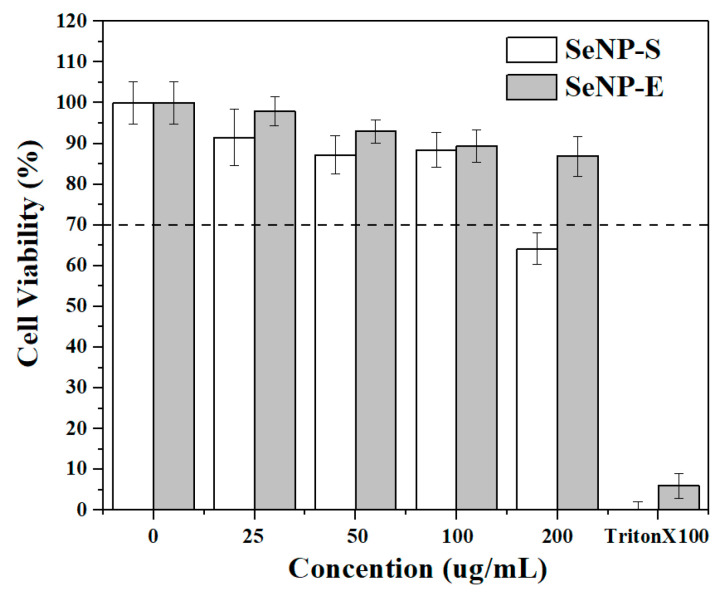
Cytotoxicity of SeNPs-E and SeNPs-S. Data were averaged across three replicates, with corresponding standard deviations.

**Table 1 ijms-23-11953-t001:** Reduction property of SeO_3_^2−^ by the strain 72, 21, and 33.

Strains	*k*	R^2^	t_1/2_ (h)	Reduction Rate of SeO_3_^2−^
72	0.1081	0.966	6.41	93.7%
33	0.1297	0.973	5.34	96.7%
21	0.1162	0.957	5.96	95.2%

**Table 2 ijms-23-11953-t002:** Inhibition rate of SeNPs-S and SeNPs-E to four pathogens at 200 μg/mL.

Sample	Inhibition Rate (%)			
	*P. aeruginosa*	*E. coli*	*S. aureus*	*L. monocytogenes*
SeNPs-S	91.1	92.1	95.2	84.5
SeNPs-E	86.7	90.8	96.8	86.3

## Data Availability

Not applicable.

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
