# Peer review of "High-Efficiency Reducing Strain for Producing Selenium Nanoparticles Isolated from Marine Sediment"

_ijms, 2022, doi:10.3390/ijms231911953_

Round 1

Reviewer 1 Report

Reviewer Comment

Comments and Suggestions for Authors

The study discussed the Selenium nanoparticles (SeNPs) .The article is interesting and may attract the readers, however, following points should be addressed before acceptance.

1-Abstract section is not well written. Rewrite the abstract section and clearly mention the objective and research statement of the study.

2-      In the citations, we see lots of old references from 1980, to 2000, ... There are many new publications in this area and I'm worried if did you check them or not?! 

3-      Make a flowchart for Methodology Adopted

4-      The dpi of figure must be atleast 300 .

5-      Write the conclusion precisely.

6-       There is not any implications section that is so important for any academic research

Reviewer 2 Report

The manuscript can be accepted as it is.

Reviewer 3 Report

In this manuscript, Quan and coworkers reported a method for synthesizing selenium nanoparticles (SeNPs) using an ocean isolated bacteria strain and its extracellular extract. The authors further evaluated the inhibition function of these SeNPs and cytotoxicity to human epithelial cells, and showed potential in applications using these biosynthesized SeNPs. The experiments in this manuscript are straightforward and well documented. The interpretations of data are comprehensive. Few minor points need to be addressed before publication in IJMS.

It is intriguing that only the extracellular but not the pericellular or intracellular extract could reduce the sodium selenite. However, it seems the three extracts may present different pH, because the extraction buffer and chemicals are very different. It is known that pH is a contributing factor to enzymatic functions and binding affinities of biomolecules. Thus, the SeNP formation location assay may not be conducted under parallel comparison. The pH of three extracts should be verified and if there are differences between groups, it is suggested to adjust the pH to physiological level of bacteria before repeating this assay. 

Please provide information about the function of 25% sucrose solution in the extracellular extract isolation process. The quality of sucrose should be confirmed, because although sucrose itself is not a reducing chemical, the degradation products of sucrose (glucose and fructose) are reducing sugars. Thus, the sucrose may or may not be an influence factor to the SeNP reduction process, which should be experimentally validated. 

Figure 2(b), it is shown that the strain 33 has closest relationship to Bacillus halotolerans, and second closest to Bacillus subtilis but not Bacillus sp. Please elaborate the interpretation that the strain 33 belongs to Bacillus sp.

A proofreading should be carried out to avoid text mistakes. Few examples, line 278, the (f) panel explanation is missing; line 335 (d) should be "Raman" spectrum.

Reviewer 4 Report

In this paper, Zhang et al. studied 40 different strains to synthesize selenium particles. They screened them in terms of reduction rate and claimed the efficiency could be increased up to 96% when strain 33 is used. Then, they characterized prepared particles in size, shape, and cytotoxicity. They compared particles prepared by whole cells (S) and extracellular extracts (E), showing S type is smaller and less toxic than E type particles.  

This study offers an interesting and useful study and proposes three strains to biologically synthesize selenium particles with a high reduction rate. However, in some parts, the authors need to bring more clarity and discuss more their results and applications. This manuscript in its version cannot be recommended for publication, and the authors need to address the following minor and major comments before considering this manuscript for publication in IJMS journal.

Major and minor comments:

-         What are the main differences between chemically and biologically synthesized selenium particles? Did the authors compare them in terms of stability, toxicity, shape and etc.? The advantages and disadvantages need to be discussed briefly in the manuscript.

-         According to SEM images, the level of aggregation is very high for both types of particles (S& E). This could impact the result of the cytotoxicity test. The high level of aggregation decreases the cellular update, leading to a lower level of cytotoxicity. How the authors addressed this problem?   

-         NPs size has been substantiated to play a significant role in the level of NPs cytotoxicity. S type is much smaller than E type and is expected to be more toxic. It would be useful if the authors could compare the cellular uptake for both S and E types.  

-         The quality of figures and graphs need to be enhanced. For example for figure1

-         Figure 1: the number of replicates should be provided.

-         Figure 5f: typo

-         Line 391: please add the relevant reference

- The numbering for figures should be modified. Jumping from 6 to 10

-         Figure 9 is missing

-         Figure 10: define the error bar in the caption and provide the number of replicates

-         Table 2: Please define inhibitory rates and explain how they were calculated.

-         Conclusion was repeated two times. In the "conclusion", please discuss the future applications of these two types of nanoparticles.

Reviewer 5 Report

Zhang et al. presented a work about the preparation of selenium nanoparticles by marine bacillus. They isolated the microbes, tested their activity in the reduction of Se(IV) to Se(0); furthermore, the authors characterized the SeNPs by UV, SEM, XRD, IR and Raman, and they tested their toxicity and their antibacterial activity. Although several articles reported the (bio)synthesis of SeNPs, the presence of different biomolecules (e.g., amino acids) on the selenium surface could affect the properties (size, solubility, compatibility, lifetime, etc.) and their uses. So, I appreciated this manuscript and the work of the authors. I suggest accepting the manuscript after minor revisions:

-          UV spectroscopy: the authors should discuss about the SPR of the selenium as a region in the range 250-300 nm rather than a single peak at 275 nm or at 284 nm (see also Srivastava, N., Mukhopadhyay, M. Bioprocess Biosyst Eng 38, 1723–1730 (2015)); the differences in the absorbance and relative explanations are just appropriate.

-          In the XRD, the JCPDS surely is the spectrum of Se(0) but it is better to clarify it

-          Line 386: there is a break in the discussion; SEM descriptions are not introduced. I suggest reformulating this initial part

-          The generic NPs are written sometimes NP or NPS, etc; NPS are written as NPS and NP-S; nano selenium is written also as nano-Se: I suggest a deep check of the text

-          The acronym SPR was not explicated; I suggest to explicit the plasmon absorbance only the first time

-          The authors should use mg with m as Greek and not ug

-          The authors should correct spectrums with spectra (Latin words)

-          p.6 line 269: the subscript and superscript in the formula

-          Fig. 1: the description in the caption is not fully clear for the reader

-          Fig. 4, caption: Roman should be changed by Raman

-          Line 338 and 341: 1640 or 1540?

-          Line 391-392: I suggest adding the reference

-          Fig. 6 and fig. 7: concention instead of concentration?

-          References: the number of the articles is not exact

Round 2

Reviewer 1 Report

Thank you for revising the manuscript as suggested and now it can be published in present form

Reviewer 4 Report

The authors improved the manuscript and addressed my comments. The manuscript can be recommended for publication in IJMS.